# *Opuntia dillenii* as a Nutraceutical and Dietary Resource for Disease Prevention and Management: A Systematic Review

**DOI:** 10.3390/nu17243915

**Published:** 2025-12-14

**Authors:** Nisa Buset-Ríos, Mussa Makran, Ruymán Santana-Farré

**Affiliations:** 1Department of Basic Health Sciences, Faculty of Health Sciences, Universidad Fernando Pessoa-Canarias, 35450 Santa María de Guía, Las Palmas, Spain; rsantana@ufpcanarias.es; 2Department of Nutrition, Faculty of Health Sciences, Universidad Fernando Pessoa-Canarias, 35450 Santa María de Guía, Las Palmas, Spain; mmakran@ufpcanarias.es

**Keywords:** health and nutrition, prickly pear, disease prevention, disease management, nutraceutical, functional food, phytochemical, bioactive compound

## Abstract

Background: Chronic diseases are leading causes of morbidity and mortality worldwide and their prevalence is increasing due to aging and lifestyle factors. A central element in their pathophysiology is chronic low-grade inflammation, linking metabolic, cardiovascular, neurodegenerative, and proliferative disorders. In this context, *Opuntia dillenii*, a cactus species traditionally used in folk medicine, has attracted considerable scientific interest due to its promising nutraceutical potential. Methods: This systematic review was conducted through a PRISMA-guided literature search using PubMed, Scopus, and Web of Science, identifying 45 studies that analyze the phytochemical composition and biological activity of *O. dillenii*. Results: The compounds highlighted include betalains, polyphenols, flavonoids, and polysaccharides that exhibit potent anti-inflammatory and analgesic effects by modulating key inflammatory mediators. In addition, *O. dillenii* demonstrates antiproliferative activity, inducing apoptosis and inhibiting tumor growth in various *in vivo* models, suggesting a potential role in cancer prevention and as a complementary therapy. The cactus also exhibits antiatherogenic and hypotensive effects, as well as hypolipidemic and antidiabetic properties by improving lipid profiles, reducing serum cholesterol and triglycerides, and enhancing insulin sensitivity. Furthermore, its protective actions against tissue damage extend its therapeutic potential. Antimicrobial properties have also been reported, reinforcing its value as a functional food. Conclusions: Taken together, the evidence supports the use of *O. dillenii* as a versatile nutraceutical resource with a low toxicity profile, capable of contributing to the prevention and treatment of various chronic inflammatory and metabolic diseases. Nevertheless, human clinical trials are needed to validate these findings and explore their full therapeutic utility.

## 1. Introduction

Chronic non-communicable diseases (NCDs), such as cardiovascular disease, type II diabetes *mellitus* (DMII), cancer, and neurodegenerative disorders, are leading causes of morbidity and mortality worldwide [1]. Their prevalence continues to rise due to population aging, sedentary lifestyles, and unhealthy dietary habits [2]. A common pathological hallmark of these conditions is the persistence of low-grade inflammation and metabolic dysregulation, which drive tissue damage and organ dysfunction. Consequently, the global public health agenda emphasizes preventive strategies targeting modifiable lifestyle factors, among which diet plays a significant role [3].

Beyond conventional pharmacological therapies, attention has been directed toward the use of diet-derived bioactive compounds as adjunctive or preventive tools against NCDs [4]. These compounds may modulate key metabolic pathways, reduce inflammation, and restore cellular metabolism. In this context, functional foods and nutraceuticals have emerged as promising resources to promote health and mitigate the onset or progression of metabolic and inflammatory disorders. Functional foods are defined as those providing health benefits beyond their basic nutritional value, while nutraceuticals refer to purified or concentrated forms of bioactive substances capable of exerting measurable physiological effects [5]. Together, these approaches form an integrative nutritional strategy that bridges food science and medicine, offering complementary tools for disease prevention and management. 

Among the wide array of plant species investigated for their nutraceutical potential, cacti of the genus *Opuntia* have attracted significant interest, particularly species such as *Opuntia ficus-indica* and *Opuntia dillenii* [6]. These plants belong to the Cactaceae family and include several prickly pear species. Native to the Americas, *Opuntia* species have been integral to the diets and traditional medicine of indigenous cultures, not only as a food source but also for the treatment of digestive issues or skin conditions. Scientific research has corroborated these traditional uses, showing that *O. dillenii* contains a variety of bioactive compounds, including polyphenols, flavonoids, betalains, and polysaccharides, that contribute to its antioxidant, anti-inflammatory, and metabolic effects [7]. Furthermore, this species has shown potential in protecting against oxidative stress, supporting liver function, and modulating metabolic pathways related to insulin resistance and lipid metabolism. The diversity of its pharmacological properties is likely due to the complex chemical profile of the plant, which varies depending on factors such as the part of the plant used, extraction method, and environmental conditions.

Antioxidant properties have historically been the most studied and are well supported by a recent systematic review [8]. Nevertheless, these effects only represent a fraction of its therapeutic potential. Thus, exploring *O. dillenii* beyond its antioxidant role is crucial to better understand its preventive and therapeutic relevance. 

Emerging evidence suggests that extracts and isolated compounds from *O. dillenii* may exert hypolipidemic, antidiabetic, anti-inflammatory, and antiproliferative effects through diverse mechanisms. Despite these promising findings, no systematic review to date has comprehensively synthesized the outcomes related to the non-antioxidant biological activities of *O. dillenii*. In this context, the present work aims to critically evaluate the biological properties of *O. dillenii*, focusing on mechanisms related to inflammation, lipid metabolism, glucose regulation, cellular proliferation, tissue protection, and antimicrobial activity. By integrating the available experimental data, this work seeks to clarify the physiological relevance of these effects, identify research gaps, and provide a comprehensive perspective on *O. dillenii* as a multifunctional nutraceutical candidate. This approach not only contributes to a broader understanding of the therapeutic potential of dietary bioactive compounds but also aligns with the global effort to develop evidence-based functional foods capable of supporting the prevention and management of NCDs. 

## 2. Methods

### 2.1. Data Source and Search Strategy

This systematic review was performed following the general principles published in the Preferred Reporting Items for Systematic Reviews and Meta-Analyses (PRISMA) 2020 guidelines [9,10]. The search was carried out in April 2025 in three commonly used databases: PubMed, Web of Science, and Scopus. The search strategy was constructed by combining all relevant terms related to *O. dillenii* in the following query: (“*Opuntia dillenii*” or “*O. dillenii*” or “*Opuntia stricta* var. *dillenii*”). No date, type of publication, or language filters were applied in order to initially include all articles related to the object of study.

### 2.2. Inclusion and Exclusion Criteria

The eligibility criteria for our systematic review were as follows: (a) *in vitro* or *in vivo* original research studies, (b) studies that referred to *O. dillenii*, and (c) studies reporting data about protective or therapeutic effects on different diseases. In order to refine our search, the exclusion criteria were as follows: (a) reviews and systematic reviews, (b) studies related only to chemical characterization, (c) studies about the combination of *O. dillenii* with other products, (d) studies related only to food technology, (e) ethnobotanical studies, (f) studies on species different from *O. dillenii*, (g) studies that only report data on antioxidant capacity, (h) studies not related to food applications, (i) congress communications, and (j) patents (Appendix A).

### 2.3. Data Extraction

Once all articles were selected, titles and abstracts were examined for their eligibility. Three researchers (M.M., N.B., and R.S.) carried out the data extraction at this stage of the systematic review. After merging the results from the three databases, 239 duplicated studies were excluded. Then, 257 studies were excluded based on the previously described exclusion criteria. Finally, this screening led to the identification of 45 studies suitable for review (Figure 1). Any disagreements regarding the application of the exclusion criteria were resolved through discussion among the three researchers.

### 2.4. Risk of Bias

For animal studies, the three reviewers (M.M., N.B., and R.S.) independently assessed the risk of bias by using the SYRCLE tool for *in vivo* research [11]. As animal intervention studies differ from randomized clinical trials in many aspects, the systematic review methodology must be adapted and optimized for these situations. SYRCLE is based on the Cochrane risk-of-bias tool for randomized human clinical trials but adjusted for aspects of bias that play a specific role in animal intervention studies. For cell-based works, they also used the same tool but adapted it to the *in vitro* assays. The information analyzed in all the articles was as follows: (a) selection bias, (b) performance bias, (c) detection bias, (d) attrition bias, (e) reporting bias, and (f) other sources of bias. For antimicrobial and enzymatic activity assays, no risk of bias was assessed due to the lack of an appropriate tool for this type of experiment. Once discussed by the three reviewers, the bias information for each study was summarized in the Appendix A where “Yes” means low risk of bias, “No” means high risk of bias, and “Unclear” means that the information reported was not enough.

### 2.5. Data Analysis

Once the 45 studies were selected, full texts were analyzed by the three researchers (M.M., N.B., and R.S.) and 5 different data extraction sheets were elaborated on. All the sheets compiled the following information: (a) biological model employed, (b) *O. dillenii* treatment applied, (c) type of assays performed or tissue damage, and (d) significant results obtained. As previously, any disagreement during this phase was resolved by the three researchers through fair discussion.

## 3. Results and Discussion

This section summarizes the experimental findings on various extracts and bioactive components of *O. dillenii*, which act as modulators of oxidation and inflammation. This contributes to the mitigation of key metabolic dysfunction processes, the reduction in tissue damage, and even revealing antimicrobial properties (Figure 2). Across the studies reviewed, there is a consistent indication of an absence of toxic effects for the different routes of administration and doses tested, suggesting a favorable safety profile under the experimental conditions.

### 3.1. Anti-Inflammatory and Analgesic Effects

*O. dillenii* has been recognized for its anti-inflammatory and analgesic properties, mainly attributed to bioactive compounds, such as betalains and polyphenols, that inhibit pro-inflammatory mediators and reduce oxidative stress. Several studies have demonstrated the ability of this cactacea to reduce inflammation and pain, positioning it as a promising natural resource for the development of complementary therapies with fewer adverse effects (see Table 1).

Ihsan-ul-Haq et al. [12] evaluated the anti-inflammatory power of the methanolic extract of the fruit. It showed no positive results when evaluating the inhibition of LPS-induced NO production in RAW 264.7 cells, suggesting little or no anti-inflammatory activity in this specific pathway. In contrast, El Hassania et al. [13] used lipoxygenase (LOX) inhibitory activity to examine the anti-inflammatory potential of seed oils, finding inhibition percentages from 81 to 93% at concentrations ranging from 20 to 100 μg/mL. The Aknari variety (AK) has the lower mean inhibitory concentration (IC_50_, 56.72 μg/mL), followed by Harmocha (HA) (74.59 μg/mL) and Imatchan (IM) (91.56 μg/mL). These findings showed moderate LOX inhibitory activity in comparison with standard quercetin (71.4 μg/mL). Anti-inflammatory properties were also evaluated using the inhibition of bovine serum albumin (BSA) denaturation. In this case, all samples (AK, HA, and IM) tested at 20–100 μg/mL showed 85–97% inhibition of BSA denaturation. IM exhibited the most potent inhibitory effect, which was statistically similar to that of diclofenac.

Another study examined different extraction conditions to improve the recovery of compounds with anti-inflammatory activity. Using ultrasound-assisted extraction (UAE), Gómez-López et al. [14] identified 5 min, 50% amplitude, 15% ethanol, and 20 °C as optimal conditions, which increased the yield of hydrophilic antioxidants such as betalains and phenolics. Under these conditions, extracts showed higher yield and bioactivity, with anti-inflammatory action (hyaluronidase inhibition) strongly correlated with flavonoid content.

Siddiqui et al. [15] showed that the methanol (MeOH) extract, its fractions T-1 (rich in opuntiol) and T-2 (rich in opuntioside), and the isolated compounds opuntiol and opuntioside, significantly reduced arachidonic acid (AA)-induced ear edema in mice in a dose-dependent manner. The inhibition of edema ranged between 36 and 65% for the MeOH extract, 33 and 50% for T-1, 19 and 33% for T-2, and 45 and 59% for both opuntiol and opuntioside. In comparison, anti-inflammatory drugs celecoxib and indomethacin reduced edema by 28–53% and 46–52%, respectively, while dexamethasone and nordihydroguaiaretic acid did not produce significant reductions. Similarly, in ear edema induced by 12-O-tetradecanoylphorbol-13-acetate (TPA), the MeOH extract, fractions T-1 and T-2, along with opuntiol and opuntioside, effectively suppressed edema. The phospholipase A2 (PLA2)-induced paw edema showed, in a dose-dependent manner, a maximum reduction at 60 min by opuntioside (66%) and opuntiol (45%). The carrageenan-induced peritonitis, used to evaluate vascular permeability, was significantly attenuated by extract, fractions, and pure compounds after the carrageenan injection. Prominently, opuntioside reduced the vascular permeability by 53% and opuntiol by 48%. These findings underline the significant anti-inflammatory potential of *O. dillenii* extracts and pure compounds, with an efficacy comparable to well established pharmacological agents.

Finally, in the hepatic steatosis model proposed by Besné-Eseverri et al. [16], steatotic rats that received either a low or a high dose of peel extract showed a significant decrease in serum C-reactive protein (CRP) compared with rats that only received the steatotic diet. These levels decreased even below those observed in the control group maintained on a standard diet. However, at hepatic level, the extract did not significantly modify the expression of inflammatory proteins related to the NLRP3 inflammasome, caspase-1, IL-1β, IL-6, or p38 MAPK activation. These findings indicate that its anti-inflammatory effect does not appear to be directly mediated by the modulation of these hepatic inflammatory pathways. Thus, the extract attenuated systemic inflammation more than local hepatic inflammation, suggesting that it could exert a relevant anti-inflammatory effect in other organs or systems not evaluated in the study.

**Table 1 nutrients-17-03915-t001:** Comparison of anti-inflammatory and analgesic effects.

Biological Model	Treatment	Assay	Result	Reference
293/NF-κB-Luc HEK cells	20 μg/mL 6 h (fruit extract)	Inhibition of TNF-α-induced NF-κB	No inhibition	[12]
RAW 264.7 cells	10 μL/mL 20 h (fruit extract)	Inhibition of LPS-induced NO production	No inhibition	
-	20–100 μg/mL	LOX inhibition test	Moderate anti-inflammatory AK < HA < IM	[13]
-	20–100 μg/mL	Inhibition of BSA denaturation	Anti-inflammatory activity IC_50_ similar to diclofenac IM > AK > HA	
Sprague-Dawley rats	50–400 mg/kg b.w. extract	Carrageenan-induced paw edema	Inhibition of edema after 3 h *O. dillenii* (400 mg/kg b.w.) = 53.25%	[17]
	50–400 mg/kg b.w. extract	Hot plate test	Reaction time *O. dillenii* (400 mg/kg b.w.) = 12.0 min	
Swiss albino mice	50–400 mg/kg b.w.	Writhing test	Writhing movements *O. dillenii* (400 mg/kg b.w.) = 20.36 ASA (70 mg/kg b.w.) = 16.93	
Albino rats	Extract (200 mg/kg b.w.) or subfractions (50 mg/kg b.w.)	Carrageenan-induced paw edema	Inhibition of edema after 4 h *O. dillenii* flowers (200 mg/kg b.w.) = 48.9% Subfraction 2 (50 mg/kg b.w.) = 53.2%	[18]
	Extract (200 mg/kg b.w.) or subfractions (50 mg/kg b.w.)	Electrical noxious stimulation to tail	Analgesic activity *O. dillenii* flowers = 83.9% Subfraction 2 = 92.2%	
NMRI mice	MeOH extract, fractions (T-1 and -2), opuntiol, and opuntioside	AA-induced ear edema	Inhibition of edema MeOH extract > opuntiol = opuntioside > T2 > T1	[15]
		TPA-induced ear edema	% Reduction ear edema T1 > opuntiol > MeOH > T2	
		PLA2-induced paw edema	% Reduction paw edema opuntioside > T2 > opuntiol > T1	
		Carrageenan-induced peritonitis	% Inhibition opuntioside > opuntiol > MeOH extract > T1 > T2	
NMRI mice	Cladodes MeOH extract, T1, T2, opuntiol, or opuntioside	AcOH-induced writhes	IC_50_ MeOH extract > T1 > T2 > opuntiol > opuntioside	[19]
		Formalin-induced paw licking response	IC_50_ late phase MeOH > T2 > T1 > opuntiol > opuntioside	
		Hot plate-induced jumping response	Latency time = 60″ opuntioside > opuntiol	
Wistar rats	Ethanolic extract (peel fruit)	Standard diet HFHF HFHF + Low doses HFHF + High doses	No significant differences in expression of NLRP3, caspase-1, IL-1β, IL-6, TNF-α, and p38 MAPK. ↓ CRP	[16]

AA: Arachidonic acid; AcOH: Acetic acid; AK: Aknari variant; ASA: Acetylsalicylic acid; CRP (C-reactive protein) HA: Harmocha variant; HFHF: High-fat high-fructose; IM: Imatchan variant; i.p: Intraperitoneal; LOX: Lipoxygenase; MeOH: Methanol; NF-κB: Nuclear factor kappa-light-chain-enhancer of activated B cells; PLA2: Phospholipase A2; TNF: Tumor necrosis factor; TPA: 12-O-tetradecanoyl-phorbol-13-acetate. The downward arrow indicates a reduction following the administration of *O. dillenii*.

*O. dillenii* shows consistent anti-inflammatory effects, but these are pathway-selective rather than broadly immunosuppressive. The fruit extract does not reduce LPS-induced NO production in macrophages. In contrast, seed oils strongly inhibit LOX and protect BSA denaturation, and opuntiol and opuntioside markedly reduced edema and vascular leakage in AA, TPA, PLA_2_, and carrageenan models. Together, these data suggest actions on AA metabolism, rather than suppressing machrophage NO signaling. Fruit and peel (rich in flavonoids and betalains) show hyaluronidase inhibition and BSA-stabilizing activity. This seems to support a complementary mechanism in which polyphenols preserve extracellular matrix integrity, reducing exudation and infiltration at inflamed tissues.

The steatosis model provides an additional nuance. Peel reduces systemic CRP to values below healthy controls, but without changes in NLPR3, caspase-1, IL-1β, IL-6, or p38 MAPK expression. This suggests actions on systemic inflammation, such as cytokine release from adipose tissue or low-grade inflammation, rather than directly modulating intrahepatic inflammatory pathways.

Tissue injuries and inflammation are key drivers of pain, triggering the release of various inflammatory mediators like eicosanoids, vasoactive amines, and cytokines. These substances contribute to pain by both increasing sensitivity and amplifying the body’s response to painful stimuli. *O. dillenii* has a long history of use in folk medicine, including for the relief of pain, though its use as analgesic was first evaluated by Loro et al. [17]. In this study, the anti-inflammatory effect was evaluated by the carrageenan-induced hind paw edema in male Sprague-Dawley rats. Pre-treatment with dry extract of the fruit significantly reduced the edema in a dose-dependent way from the dose 100 mg/kg body weight (b.w.), 2 h after carrageenan injection. This effect reached the maximal inhibition after 3 h with the dose 400 mg/kg b.w. (53.25%), very close to the effect of indomethacin (59.74% of inhibition with 5 mg/kg b.w.). In a hot plate test, *O. dillenii* extract significantly increased the reaction time of the rat, starting at 100 mg/kg b.w. (22.37%) in a dose-dependent manner, reaching the maximum effect (96.30%) at 400 mg/kg b.w. at 15 min after injection and then slowly decreasing. Morphine (2 mg/kg b.w.), used as the reference drug, significantly increased the reaction time, reaching its maximum 1 h after treatment. The authors suggested central analgesic properties potentially linked to increased GABA levels. In addition, this initial research tested a lyophilized aqueous extract from the fruit in male Swiss albino mice subjected to the acetic acid (AcOH) writhing test. At intraperitoneal (i.p.) doses of 50–400 mg/kg b.w., the aqueous extract inhibited the writhing responses of mice, showing a number of writhes significantly lower than the control group. The maximal inhibition of the nociceptive response was 49.65% with the dose of 400 mg/kg b.w. However, acetylsalicylic acid (ASA) exerted a greater effect, inducing a protection of 58.14% at a dose of 70 mg/kg b.w.

Continuing with the study of the analgesic effect, Ahmed et al. [18] applied electrical stimulation to male albino rats’ tails. The rats were treated with alcoholic extracts of flowers, fruits, and stems. The analgesic effect was pronounced in the alcohol extract of the fresh flowers (200 mg/kg/b.w.), compared with the effect of Novalgin (50 mg/kg/b.w.), on electric current painful stimuli. In order to characterize the active principles, the alcohol extract of the flowers was fractionated and eluted with MeOH and ethyl acetate (EtOAc). Interestingly, it showed a potent analgesic activity at a much lower dose (50 mg/kg b.w). Kaempferol 3-O-α-arabinoside, isorhamnetin-3-O-β-D-glucopyranoside, and isorhamnetin-3-O-β-D-rutinoside were isolated from these active subfractions. The results suggest that the identified flavonoids may contribute to the observed analgesic effect.

Siddiqui et al. [19] evaluated analgesic activity from cladodes MeOH extract, using chemical (AA-induced writhing and formalin-induced paw licking test) and thermal (hot plate test) stimuli. Orally administered fractions T-1 and T-2, along with the isolated compounds of opuntiol and opuntioside, elicited a significant dose-dependent reduction in abdominal writhes compared to control. These samples were highly effective in alleviating pain, showing a potency comparable to diclofenac sodium and β-sitosterol. Opuntiol exhibited a response similar to that induced by ASA. *O. dillenii*-derived test agents reduced peripheral and central pain phases, but a more pronounced effect was detectable in the second phase of paw licking. In this case, opuntioside was six-fold less effective than morphine but similar to β-sitosterol, diclofenac sodium, and indomethacin. With respect to hot the plate test, *O. dillenii*-derived test agents suppressed the centrally mediated pain with maximum analgesic response (~70%) at 60 min followed by gradual decline in latency time. Consistently, the effect of opuntioside was most pronounced and comparable to β-sitosterol with a four-fold better anti-nociceptive effect than opuntiol. These findings suggest that *O. dillenii*-derived test agents possess peripheral effects, possibly via the inhibition of inflammatory mediators, as well as central analgesic activities, more likely due to the involvement of opioid receptors. The authors suggest that COX-2 or GABAergic systems could be involved in opioid-independent mechanisms.

*O. dillenii* exhibits analgesic effects consistent with peripheral and central pathways. The differences between the extracts and the reference drugs probably reflect variations in compositions and pharmacokinetics. The studies analyzed suggest pain relief by the inhibition of inflammatory mediators, reducing peripheral sensitization, and modulating central via GABAergic inhibition and partial activation of opiod receptors, as suggested by the hot plate and antagonism studies.

### 3.2. Antiproliferative Effects

The possible antiproliferative effect of *O. dillenii* is mainly attributed to its phenolic compounds and betalains, which could induce apoptosis and modulate inflammatory pathways related to the uncontrolled growth of tumor cells. Some pharmacological studies have reported the antitumor activities of extracts from this plant, suggesting its usefulness in complementary therapies and in the development of new pharmaceutical agents with a low toxicity profile (see Table 2).

Ihsan-ul-Haq et al. [12] evaluated the chemopreventive and cytotoxic potential of extracts and fractions derived from the fruit, prepared by maceration in MeOH and subsequent fractionation by solvent–solvent extraction and column chromatography. The samples were subjected to *in vitro* assays to determine their ability to inhibit the TNF-α-induced activation of nuclear factor kappa-light-chain-enhancer of activated B cells transcription factor (NF-κB), aromatase activity, induction of quinone reductase 1 (QR1), and activation of the retinoid X receptor (RXR). The antiproliferative effect on breast (MCF-7, MDA-MB-231) and lung (LU-1) cancer cell lines was also evaluated. None of the samples showed significant activity in any of these parameters, indicating no detectable chemopreventive or cytotoxic potential in the tested models.

Li et al. [20] found that *O. dillenii* polysaccharides inhibited the proliferation of SK-MES-1 lung carcinoma cells in a dose- and time-dependent manner (24–48 h). Treated cells showed morphological alterations, S-phase arrest, and increased apoptosis (up to 31.15%). Protein expression analysis revealed significant upregulation of p53 (3.83–1.99 fold) and PTEN (phosphatase and tension homolog deleted on chromosome ten) (3.59–2.61-fold), confirming cytotoxic and pro-apoptotic effects.

Pavitra et al. [21] reported that pectin, pulp, and betalains from *O. dillenii* significantly inhibited the viability of Ehrlich ascites carcinoma (EAC) cells (cell model of hepatocellular carcinoma). Betalains were the most active (IC_50_ = 71.9 µg/mL at 48 h in MTT assay), inducing 78.88% apoptosis, while pectin and pulp induced 39% and 38%, respectively. Microscopy and TUNEL analyses confirmed apoptosis via DNA fragmentation. In normal NIH3T3 cells, cytotoxicity was minimal, and all fractions showed < 5% hemolysis, indicating good biocompatibility and a favorable safety profile.

With regard to *in vivo* studies, Ejaz et al. [22] evaluated the angiogenic potential of an aqueous extract of cladodes, which at different concentrations (4–10%) exhibited a marked dose-dependent antiangiogenic response in the chicken chorioallantoic membrane (CAM) model. A significant decrease in the total area of the CAM, and in the diameter of the primary, secondary, and tertiary blood vessels, was observed in all groups treated with *O. dillenii* extract compared to the control group. Three-dimensional evaluation, using computerized imaging, showed a reduction in the proliferation, migration, and differentiation of blood vessels, as well as damage to the angular pattern and height of the vasculature, reflecting significant structural deterioration in the vessels formed after treatment with *O. dillenii*. Histological analysis showed thinning of the ectodermal layer, decreased capillary plexus formation, damage to the extracellular matrix, and reduced fibroblasts in the vascular regions of the CAM. Severe damage to secondary and tertiary vessels and poor vascular differentiation and proliferation were also observed. This study provides evidence that *O. dillenii* extract has considerable antiangiogenic potential, suggesting that its phytochemicals could be a source of new complementary or preventive treatments in pathologies mediated by abnormal angiogenesis or in cancer.

The studies analyzed point to an antitumor potential of *O. dillenii* depending on the type of extract, the cell model, and the targets tested. Methanolic fruit extract shows no chemopreventive activity. In contrast, polysaccharides inhibit proliferation and promote apoptosis and p53 and PTEN expression. Similarly, pectin, pulp, and betalains reduce EAC cell viability, with minimal toxicity in normal cells, indicating a safe profile. On the other hand, the aqueous extract of cladodes shows antiangiogenic action. The apparent discrepancy between inactive extracts and highly effective fractions could be due to differences in composition and targets: negative results focus on specific regulatory pathways, while polysaccharides and hydrophilic pigments have been tested on more general cell cycle nodes. Taken together, these results suggest a clinical potential for *O. dillenii* that resides in the combination of tumor suppressor activation, selective redox stress, and vascular support blockade, which should be confirmed by comparative studies in more lines and animal models. Also, they support its further exploration as a source for novel complementary therapies or as a basis for new pharmaceutical development targeting neoplasias and pathological angiogenesis. Nevertheless, the variable outcomes across studies highlight the need for standardized extract preparation, comprenhensive mechanistic research, and robust *in vivo* validation.

### 3.3. Anti-Lipogenic and Lipid-Lowering Effects

*O. dillenii* has been the subject of numerous experimental evaluations exploring its effects on lipid metabolism and adipogenesis. Various extracts, including whole fruit, peel, pulp, and industrial by-products, have demonstrated modulatory effects on lipogenesis, fatty acids uptake, and lipolysis in both cellular and animal models (see Table 3).

The study by Gómez-López et al. [23] suggests that extracts from whole fruit, peel, and pulp were effective in reducing triglyceride (TG) accumulation in mature 3T3-L1 adipocytes. At a concentration of 50 μg/mL, pulp extract was the most effective. In contrast, at a concentration of 100 μg/mL, the efficacy of both whole fruit and pulp were similar. To elucidate the mechanism of action, the expression of genes and proteins involved in 3T3-L1 mature adipocyte metabolism were evaluated, revealing a significant reduction in CD36 observed in samples treated with pulp extract. This implies a lower availability of fatty acids and, therefore, a reduction in TG in the adipocytes. Conversely, phosphorylated hormone-sensitive lipase (HSL) increased in samples treated with peel or pulp extract, which correlates with activation of the lipolytic pathway. Altogether, these results indicate that the pulp extract exhibited a dual effect by reducing de novo lipogenesis and the uptake of blood fatty acids, while simultaneously increasing lipolysis. In a further study, Gómez et al. [24] evaluated the action of prickly pear extracts at 10–100 μg/mL on 3T3-L1 pre-adipocyte differentiation. All extracts led to an increase in both TG accumulation and cell number. Analysis of adiponectin and the expression of genes involved in the adipogenic process revealed that incubation with the whole fruit extract led to upregulation in the expression of C/EBP-β and a tendency towards an elevated value of SREBF-1. Unlike Li et al. [25], Gómez-López et al. did not observe changes in the expression of PPARγ in the adipocytes. Based on these results, they concluded that all the tested extracts are able to stimulate the first phase of adipogenesis.

Besné-Eseverri et al. [26] tested the effect of whole fruit, peel, pulp, and the industrial by-product on murine AML12 and human HepG2 hepatocytes treated with palmitic acid. They observed a significant reduction in TG accumulation in AML12 cells treated with whole fruit (100 μg/mL), but only a trend toward reduction in HepG2 cells, where the pulp treatment (100 μg/mL) appeared most effective. To determine the processes responsible for the lipid-decreasing effect, the expression of proteins implicated in lipid metabolism was measured, but none of the *O. dillenii* extracts modified fatty acid synthase, FTA2, or DGAT2 protein levels. In contrast, a downregulation of CD36 expression was observed, consistent with Gómez-López’s findings in 3T3-L1 cells [23].

The study by Serrano-Sandoval et al. [27] confirmed the effect of *O. dillenii* on reducing liver lipids and its potential to prevent hepatic steatosis. Fractionation of the fruit extract identified phenolic acids and betalains as the main active compounds. Fractions 1, 9, and 10 significantly reduced intracellular TG levels in HepG2 cells (−37% to −74%), with the strongest effect observed for pool 1, rich in quinic and piscidic acids. These findings indicate that both phenolic compounds and betalains contribute to the modulation of lipid metabolism, highlighting their combined therapeutic potential against hepatic steatosis and dyslipidemia.

Zhao et al. [28] administered *O. dillenii* polysaccharide-Ia (ODP-Ia), obtained from aqueous extracts of cladodes, intragastrically at 100, 200, and 400 mg/kg/ b.w. for 28 days, to Sprague-Dawley rats fed with high-fat emulsion. They observed a significant decrease in serum lipid levels and an increase in serum high-density lipoprotein (HDL) cholesterol level in hyperlipidemic rats. Moreover, the ODP-Ia administration significantly increased serum lecithin–cholesterol acyltransferase activity (LCAT), modulating cholesterol metabolism.

Bouhrim et al. [29] utilized a murine model of CCl_4_-induced liver injury in Wistar rats. They treated the animals with *O. dillenii* essential oil (ODSO), extracted from seeds, at a dosage of 2 mL/kg b.w./day for one week prior to the initial CCl_4_ administration. The treatment was continued for an additional week before the second CCl_4_ exposure. This regimen resulted in a reduction in TG and very-low-density lipoproteins (VLDLs), alongside an increase in glycemia and HDL. Bouhrim et al. [30] also demonstrated this hypolipidemic effect in high-fat diet-fed Swiss albino mice. They observed that the administration of ODSO resulted in a reduction in total cholesterol (TC), TG (26%), and atherogenic index. Furthermore, they noted an increased percentage ratio of HDL to TC (45%), without a significant effect on absolute HDL levels. To confirm the compounds responsible for this metabolic improvement, Bouhrim et al. [30] evaluated the chemical profile of ODSO, revealing a high content of phenolic compounds.

Recently, El-Sofany et al. [31] studied the effects of betanin administration on Wistar rats with obesity and DMII induced by a high-fat high-fructose (HFHF) diet for 90 days. They observed that betanin, at a dose of 60 mg/kg b.w. per day, led to a reduction in lipase activity or secretion in the intestine (52%), serum (37%), and pancreas (51%), accompanied by a significant reduction in b.w. (30%). Furthermore, the oral administration of betanin reduced the levels of serum TC and LDL-C, and increased HDL-C, bringing the values closer to those observed in normal rats.

**Table 3 nutrients-17-03915-t003:** Comparison of anti-lipogenic and lipid-lowering effects.

Biological Model	Treatment	Assay	Results	Reference
3T3-L1 mature	Whole fruit, peel, pulp (10–100 μg/mL)	Measurements of TG RT-PCR WB	(−) De novo lipogenesis (+) Lipolytic pathway ↓ TG, CD36 ↑ HSL	[23]
3T3-L1 pre-adipocytes	Whole fruit, peel, pulp, bagasse (10–100 μg/mL)	Measurements of TG RT-PCR	(+) lipid accumulation ↑ TG, C/EBP-β, and adiponectin	[24]
AML12	Whole fruit (100 μg/mL) Peel (10 μg/mL)	Measurements of TG RT-PCR	↓ TG, CD36	[26]
HepG2	Peel, pulp, and bagasse extracts (100 μg/mL)	RT-PCR	↓ CD36	
HepG2	Fruit extract 48 h (25 μg/mL) (quinic and piscidic acid/betalains)	Oil red O TG quantification	↓ TG ↓ Hepatic lipids ↓ Total fats ↓ Lipogenesis	[27]
Sprague-Dawley rats	ODP-I (cladode aqueous extract) 100–400 mg/kg b.w. HFD	Measurement of serum lipid profiles and LCAT activity	↓ Lipid level ↑ HDL ↑ LCAT ↓ LDL	[28]
Wistar rats	ODSO (2 mL/kg/day) before CCl_4_	Serum biochemical determination	Recover HDL, LDL, and TC levels	[29]
Swiss albino mice	ODSO HFD	Serum biochemical determination	↓ TG ↓ TC ↑ HDL/TC	[30]
Wistar rats	Betanin from pulp (60 mg/kg b.w.) (DMII HFHF-induced)	Determination of lipase activity/Biochemical analysis	(−) Lipase activity ↓ Enzyme secretion ↓ 30% in b.w. ↓ TC and LDL-C ↑ HDL-C	[31]

b.w.: Body weight; C/EBP-β: CCAAT-enhancer-binding proteins β; CD36: Cluster of differentiation 36; DMII: Type 2 diabetes *mellitus*; HDL: High density lipoprotein; HFD: High-fat diet; HFHF: High-fat high-fructose diet; HSL: Hormone-sensitive lipase; LCAT: Lecithin–cholesterol acyltransferase activity; LDL: Low-density lipoprotein; ODP: *O. dillenii* polysaccharide; ODSO: *O. dillenii* seed oil; RT-PCR: Real-time polymerase chain reaction; TC: Total cholesterol; TG: Triglyceride; WB: Western blot. The arrows (up or down) indicate increases or decreases in biomarkers resulting from the administration of *O. dillenii*.

In summary, the available evidence demonstrates that *O. dillenii* exerts a significant regulatory influence on lipid homeostasis. The stimulation of adipogenesis in preadipocytes versus the reduction in TG in adipocytes and liver are better understood when considering that *O. dillenii* could act on several levels of lipid metabolism and at different stages of cell biology. In mature adipocytes and hepatocytes, a central target, CD36, reduces the flow of fatty acids into the cells, together with HSL activation and TG mobilization, limiting adipocyte hypertrophy and hepatic steatosis. In preadipocytes, early modulation of differentiation towards small and insulin-sensitive adipocytes reduces lipid overflow to the liver and muscle. At the systemic level, polysaccharides that improve the lipoprotein profile, seed oils rich in polyunsaturated fatty acids, and phytosterols that reduce hepatic cholesterol synthesis improve LDL/VLDL clearance, and may modulate nuclear receptor like PPARα. In the digestive–pancreatic axis, betanin inhibits lipases and reduces intestinal fat intake, decreasing the lipid load on the liver, which translates into lower b.w. and a better lipid profile. From this perspective, the studies are complementary: phenolic acids, betalains, seed oils, and polysaccharides act in a coordinated manner on the uptake, synthesis, transport, and absorption of lipids. These findings highlight the potential of *O. dillenii* metabolites as promising nutraceuticals for the management of hepatic steatosis, dyslipidemia, and metabolic syndrome.

### 3.4. Antidiabetic Effects

Several experimental models, *in vivo* and *in vitro*, have shown that this cactus reduces blood glucose and modulates key targets in glucose metabolism. These findings emphasize the potential of its bioactive compounds as a complementary approach for DM management (see Table 4).

The first study on the hypoglycemic effect of *O. dillenii* was carried out by Perfumi et al. [32], which validated the traditional use of fresh filtered juice. To this end, single and repeated oral doses (5 mL/kg b.w.) were administered to normoglycemic and alloxan-induced diabetic New Zealand male rabbits. Then, they evaluated blood glucose levels and oral glucose tolerance, and compared them with the effect of tolbutamide. Single or repeated doses of the juice significantly reduced hyperglycemia induced by oral glucose loading in normal and diabetic rabbits, without altering basal glucose levels. The effect is comparable to tolbutamide in normoglycemic rabbits; however, it does not increase plasma insulin levels, suggesting the existence of a possible insulin-like active compound. It also has no effect when glucose is administered intravenously, so it probably exerts its antihyperglycemic effect mainly by reducing intestinal glucose absorption, although the study does not identify the exact active components and mechanisms.

Díaz-Medina et al. [33] extended the study of the antihyperglycemic potential of *O. dillenii* to Sprague-Dawley rats, which were administered fruit pad and pulp extracts to evaluate their effects on blood glucose concentration and the glycemic curve. To test the hypothesis that the high Cr(III) content might be responsible for the cactus’ antihyperglycemic effect, the treatment with *O. dillenii* extract was compared with the administration of Cr(III) over an 8-day period. No statistically significant differences were observed among the remaining groups. Therefore, the intake of cactus pad extract resulted in a slight reduction in fasting blood glucose levels, maintaining glucose within normal ranges.

Zhao et al. [34] studied ODP-Ia, showing significant effects in streptozotocin (STZ)-induced DM in Chinese Kunming mice. Oral administration of ODP-Ia for three weeks resulted in a significant reduction in blood glucose levels (up to 53% at high doses), along with improvements in metabolic parameters such as TC, TG, HDL-C, and b.w. The proposed mechanism for the antihyperglycemic effect of ODP-Ia does not involve direct stimulation of insulin secretion. They hypothesized that it is instead rather related to liver protection and improved cellular sensitivity and response to insulin, as well as due to the regulation of key enzymes such as glucose-6-phosphatase.

Gao et al. [35] performed a similar study investigating the protective and antidiabetic action of a new water-soluble polysaccharide fraction (ODFP) extracted from *O. dillenii* fruits in STZ-induced diabetic Sprague-Dawley rats. A polysaccharide of 6479 kDa, composed mainly of rhamnose, xylose, mannose, and glucose, was isolated and purified. Oral administration of ODFP to diabetic rats for 4 weeks significantly reduced blood glucose levels, as well as excessive food and water consumption, urine production, and increased b.w. Histopathological examination showed that ODFP markedly improved the structure integrity of pancreatic islet.

The antidiabetic action in Wistar rats with STZ-induced DM was continued by Bouhrim et al. [36], but with the administration of ODSO. The seed oils significantly reduced fasting blood glucose, glycosuria, TC, TG, and liver enzyme levels aspartate aminotransferase (AST) and alanine aminotransferase (ALT), as well as improved hepatic glycogen storage. It also increased food intake and decreased b.w. loss and urinary volume in diabetic rats, without affecting water intake. The effects were dose-dependent (1–2 mL/kg b.w./day) and comparable to those observed with glibenclamide. The results indicate improved insulin sensitivity and hepatorenal protection, probably attributed to the bioactive compounds present in the oil, such as unsaturated fatty acids (linoleic, oleic), phytosterols (β-sitosterol), and phenolic compounds with antioxidant properties.

The antioxidant action of these compounds helps to mitigate oxidative stress and metabolic alterations characteristic of DMII and dyslipidemia, but further research into the mechanism of action is still needed. The same group [37] continued trials in diabetic rats induced with STZ using an oral glucose tolerance test, confirming that ODSO significantly reduces postprandial hyperglycemia in both normal and diabetic rats, without causing hypoglycemia in healthy fasting rats. They also verified the inhibition of intestinal glucose absorption and the blocking of the sodium–glucose cotransporter 1 (IC_50_ = 60.24 µg/mL), as already outlined in the study by Perfumi et al. [32]. On the other hand, a significant inhibition of α-glucosidase (IC_50_ = 278 µg/mL) and α-amylase (IC_50_ = 0.81 mg/mL) was observed, thus reducing carbohydrate degradation. Furthermore, in rats overloaded with sucrose or starch, the oil decreased postprandial blood glucose levels. Finally, Bouhrim et al. [38] verified the effect of ODSO on peripheral glucose absorption in skeletal muscle (rat hemidiaphragm), observing a significant stimulation of glucose uptake (53 mg/g/h versus 27 mg/g/h in controls). This effect was enhanced in the presence of insulin (90 mg/g/h).

El-Sofany et al. [31] investigated, for the first time, the impact of betanin on carbohydrate-digesting enzymes in rats on an HFHF diet. They demonstrated that betanin extracted from the pulp and administered daily to rats at doses of 15, 30, and 60 mg/kg b.w., via gastric gavage for 90 days, resulted in a 49% reduction in blood glucose levels. This reduction was associated with decreased activities of intestinal α-amylase, maltase, and sucrase. The results showed that betanin exerts significant inhibitory effects on α-amylase and α-glucosidase, with IC_50_ values of 63 μg/mL and 37 μg/mL, respectively. These effects could be attributed to the strong interactions between betanin and these digestive enzymes, leading to the formation of stable complexes. This could explain the potent and sustained inhibition observed. In addition, the betanin administration improved glucose catabolism in the liver and muscles, increased glucokinase activity and glycogen levels, and reduced glucose-6-phosphatase activity.

The apparent contrast between the modest effects in healthy animals and the more conclusive results in diabetic models could be explained by the pathophysiological context, treatment duration, and the type of extract. In normoglycaemic animals, the margin for improvement in basal blood glucose is limited and the effects mainly involve the attenuation of postprandial peaks. By contrast, in models with established hyperglycaemia, insulin resistance, and pancreatic damage, the combination of a lower intestinal glucose load, more efficient hepatic and muscular metabolism, and some preservation of the islets results in clear reductions in blood glucose and a broader correction of metabolic alterations. Overall, these findings indicate the notion that *O. dillenii* exerts a multifactorial antihyperglycemic action, with different fractions (juice, polysaccharides, seed oils, betalains) acting in a complementary manner on carbohydrate digestion and absorption, tissue glucose metabolism, and pancreatic integrity. This integrated profile underscores its potential as an adjunctive strategy in the management of DMII.

### 3.5. Cardiovascular Protection

Cholesterol homeostasis in peripheral cells, particularly macrophages, is essential for maintaining overall metabolic health. If cholesterol uptake exceeds efflux, these cells become foam cells or undergo apoptosis, which initiates pathological events. Efficient cholesterol efflux mechanisms are critical for preventing lipid overload in macrophages and protecting vascular health.

Li et al. [25] investigated the effect of ODP-Ia in THP-1 foam cells and showed that it acts via the PPARγ-LXRα signaling pathway. At an optimal dose of 15 nmol/L, ODP-Ia enhanced apoA-I-mediated cholesterol efflux and reduced TC content, suggesting potential benefits for the treatment of atherosclerosis. ODP-Ia significantly increased the expression of genes and proteins involved in cholesterol efflux: ABCA1, ABCG1, SR-BI, PPARγ, PPARα, and LXRα in ox-LDL-treated foam cells. This increase was dose-dependent, although it did not reach the magnitude of the positive control group (ezetimibe). Specific antagonists of LXRα (GGPP) and PPARγ (GW9662) inhibited the promotion of cholesterol efflux mediated by ODP-Ia and significantly reduced the ODP-Ia-induced expression of ABCA1, ABCG1, PPARγ, and LXRα at mRNA and protein levels. These findings confirm that ODP-Ia acts through this signaling pathway.

Saleem et al. [39] showed that the intravenous administration of the methanolic extract of *O. dillenii* cladodes (OM) elicited a reduction in arterial blood pressure in normotensive rats. It induced decreases of 28% and 54% in mean arterial blood pressure (MABP) at dosages of 1 and 10 mg/kg b.w., respectively. The hypotensive effect of the lower dosage was transient, lasting less than one minute, whereas the effect of the higher dosage persisted for a comparatively extended period of 37 min. Fractionation of the extract via preparative thin-layer chromatography yielded four bands (OM-1 to OM-4) exhibiting varying degrees of hypotensive potency. Band OM-1 induced the greatest reduction in MABP (62%) at 10 mg/kg, although at higher dosages (30 mg/kg), this effect diminished. Results suggest that opuntiol, present in band OM-1, may be responsible for the hypotensive activity observed. Lastly, band OM-2, identified as opuntioside-I, caused a 25% reduction in MABP at a dose of 3 mg/kg b.w., and exhibited comparable activity at higher doses, with a diminished effect analogous to that of atropine.

These two lines of evidence address different levels and time scales of cardiovascular regulation. The foam cell data speak to a potential role of ODP-1a in the long-term protection against lipid accumulation in the vascular wall, whereas the blood pressure experiments reveal that the methanolic extract of cladodes can acutely influence vascular tone.

### 3.6. Tissular Protective Effects

There are many agents that can cause damage to different tissues. These damages range from injuries and alterations, caused by heavy metals, to lesions provoked by drug treatments, as well as organ dysfunctions due to poor eating habits. This is the reason why some authors have focused their research on the possible preventive and therapeutic role of *O. dillenii* on a wide variety of *in vivo* tissue damage models (see Table 5).

#### 3.6.1. Hepatoprotective Effects

The liver is the primary organ responsible for xenobiotic metabolism and detoxification, making it particularly vulnerable to damage caused by exposure to toxic agents such as heavy metals, drugs, and dietary factors. Therefore, the identification of natural compounds with hepatoprotective activity capable of counteracting these insults is of great relevance for preventing or mitigating liver damage of diverse etiologies.

Ahmad et al. [40] generated a cadmium (Cd)-induced toxicity mice model to test the hepatoprotective activity of *O. dillenii*. Cd ions (50 ppm) were given in drinking water to the animals for 15 days. Then, 0.2 mL of fruit extract was administered daily by gavage on days 16 to 22. This study focused on histopathology where the Cd group revealed a misalignment of the hepatic cords and swollen hepatocytes, causing a complete obliteration of the sinusoidal spaces. The nuclei of the hepatocytes were enlarged, and their cytoplasms contained vacuolation. Kupffer cells and bi-nucleated hepatocytes were rarely visible. The presence of cellular debris was noticed near the centrilobular veins. In contrast, the livers from the animals treated with *O. dillenii* after the Cd intoxication showed signs of hepatolobular regeneration, such as the presence of oval progenitor cells and juvenile mono- and bi-nucleated hepatocytes. Also, the hepatocytes showed centrally placed compact rounded nuclei and no signs of cytoplasm vacuolation. However, the most interesting finding was the presence of localized clumps of small undifferentiated hepatoblastic progenitor cells. The mean cross-sectional areas of the central lobular vein, hepatocyte, and hepatocytic nucleus were significantly higher in the Cd group (6368.34 µ^2^, 400.9 µ^2^ and 82.64 µ^2^, respectively) than the Cd + fruit extract group (3824.64 µ^2^, 389.51 µ^2^ and 76.14 µ^2^, respectively). Additionally, the mean number of hepatocytes per hepatic cord was significantly higher in Cd + fruit extract (10.84) than in the Cd groups (8.56). Furthermore, the mean number of progenitor cells per unit area was significantly higher in the Cd + fruit extract group (6.76) than in the control Cd (2.32) group.

Interestingly, another research group studied the hepatoprotective effects of *O. dillenii* in a similar model of Cd-induced liver injury in mice [41]. Beyond the liver enlargement found in all the damage models, the Cd administration caused hepatocyte swelling, nuclear exposure, central venous congestion, apoptosis, and inflammatory cells infiltration. The administration of ODP extract (50, 100, 200, 400, or 600 mg/kg b.w.) 14 days after the onset of the Cd damage seemed to partially relieve these pathological changes, but some of the structures were still deformed. However, with the increase in the duration of the administration period, all the negative changes were gradually alleviated. In addition to the histological findings, for Liu et al. [41] the Cd-induced liver damage model caused an increase in AST (45.6–52%) and in ALT (26.6–31.3%) serum levels, two widely used markers of liver function damage, since week two to week five. Also, it caused an elevation of the alkaline phosphatase (ALP) (38.2–43.1%), a well-known marker of cholestasis. The administration of ODP extract to the Cd group restored the serum levels of AST, ALT, and ALP to normal values, showing a good dose–effect relationship. Also, after 28 days of administration of a 200 mg/kg b.w. dose, all pathological indicators were close to normal values. This effect was similar from week two to the end of the intervention at week five, which suggested that ODP could play a continuous protective or therapeutic role in Cd-induced liver injury in mice.

The study of Shirazinia et al. [42] used a different heavy metal for the *in vivo* hepatic toxicity model: lead (Pb). In this case, they evaluated the hepatoprotective effect of *O. dillenii* fruit alcoholic extract against Pb-induced toxicity (25 mg/kg b.w./day i.p.) for a 10-day experiment. Similarly to the Cd effects previously discussed, Pb significantly increased the serum levels of ALT, AST, and ALP and liver histopathological scores (degenerated liver cords and fatty cage, hemorrhage, sinusoidal dilatation, vacuolation, and pyknotic nuclei). *O. dillenii* (100 and 200 mg/kg b.w./day) markedly reduced these enzyme levels, as well as histopathological scores (in this case, at the higher dose).

Bouhrim et al. studied the hepatoprotective effects of ODSO on a CCl_4_ acute liver damage in rats [29]. In this case, the animals were treated orally with ODSO (2 mL/kg b.w.) daily for one week before the first intraperitoneal injection of CCl_4_. Thereafter, the administration of the oil was continued for 7 days until a second injection of CCl_4_ was performed. As expected, significant increases in the serum levels of ALT, AST, and ALP in the rats were detected. The administration of ODSO significantly attenuated the elevation of these parameters. The elevation of bilirubin concentration is a sign of the dysfunction of the excretory capacity of hepatic cells. In the CCl_4_ group, a marked elevation in the plasma levels of bilirubin was detected (direct and total). Once again, the administration of the seed oils notably corrected this anomaly.

Another model of hepatotoxicity used an overdose of acetaminophen, commonly known as Paracetamol (PCM) [43]. Rats were orally treated with the fresh fruit juice of *O. dillenii* (FJOD) (2.5 or 5 mL/kg b.w.) for 7 days and then intoxicated on the fifth day with PCM (2 g/kg b.w. orally). The rats showed a significant increase in the serum levels of hepatospecific enzymes such as glutamic-oxaloacetic transaminase (SGOT), serum glutamic pyruvic transamina (SGPT), and ALP, as well as in bilirubin (direct and total), in comparison to normal control animals. Animals pre-treated with FJOD at both doses showed significant and dose-dependent correction of all these markers. In relation to the histopathological score, PCM caused focal hemorrhage, inflammation, centrilobular necrosis and degeneration, congestion of portal veins, vacuolization of cytoplasm, dilatation and congestion of sinusoids, and degeneration of hepatocytes. The FJOD pre-treated group, at both doses, showed again a dose-dependent protection against PCM damage as evidenced by very mild focal areas of necrosis, minimal centrilobular necrosis, and reduced hepatocytes degeneration.

Metabolic dysfunction-associated fatty liver disease causes a wide spectrum of alterations in the tissue. With this in perspective, Besné-Eseverri et al. [16] generated an *in vivo* steatosis rat model by the administration of a HFHF diet. At the end of the experimental period (8 weeks), in comparison with control animals, the steatotic rats presented higher levels of hepatic TG (50.4 vs. 20.0 mg/g tissue) and raised serum levels of ALT (100.1 vs. 14.1 U/L) and AST (118.2 vs. 75.8 U/L), commonly used markers of hepatic dysfunction. The daily oral administration of aqueous extract from the peel of *O. dillenii* at low or high (25 or 100 mg/kg b.w.) doses showed no significant effect in hepatic TG content (46.8 or 47.0 vs. 50.4 mg/g tissue) and the serum levels of ALT (110.0 or 86.7 vs. 100.1 U/L) and AST (129.0 or 112.4 vs. 118.2 U/L), in comparison to HFHF animals.

El-Sofany et al. [31] evaluated the hepatoprotective effect of purified betalains from *O. dillenii* pulp in obese rats with DMII induced by a HFHF diet. The HFHF group showed marked hepatic inflammation with leukocyte infiltration and elevated serum enzyme levels, glutamic-oxaloacetic transaminase (GOT) (58%) and glutamic pyruvic transaminase (GPT) (94%), and direct (96%) and total bilirubin (72%), compared to normal rats. Treatment with betalain (13, 30, and 60 mg/kg b.w. daily for 3 months) markedly reduced hepatic inflammation, lymphocyte clustering, and necrosis. Enzyme and bilirubin levels decreased to values comparable to the control group, confirming a strong hepatoprotective effect of betalains against diet-induced liver injury.

Taken together, these results reveal a clear protective role of *O. dillenii* on liver damage, independently of the source of the initial injuries. The altered levels of hepatic enzymes in serum, markers of liver dysfunction, were corrected [5,17,19,28,29,30], as well as the direct and total bilirubin levels [29,31,43]. The liver histopathological score (characterized by degenerated liver cords, hemorrhage, sinusoidal dilatation, cytoplasm vacuolation, and pyknotic nuclei, among other alterations) were also improved [31,40,41,42,43]. Finally, despite the fact that there is not an agreement on the component of the cactae responsible for the cited protective effects, isolated polysaccharides [41] and betalain [31] seem to be good candidates. Nevertheless, whole fruit [40,42,43], seed oils [29], and peel [16] exhibited promising results, so it is possible that other components are involved in the protective effects of *O. dillenii*. Despite these findings, it is essential to test the effects exhibited beyond the animal models to clarify the mechanisms behind them. Also, it will be necessary to explore which specific components of the plant are directly responsible for its protective and therapeutic properties.

#### 3.6.2. Digestive Tract Effects

To test the protective effects of FJOD on the colon, Babitha et al. [44] generated an AcOH–induced ulcerative colitis model in rats. The rats received FJOD (2.5 and 5 mL/kg, orally) daily. On the eighth day, AcOH (2 mL of 4% *v*/*v*) was administered intra-rectally to generate the damage and then the FJOD treatment were extended for 11 more days. The colitic rats showed significant problems with stool consistency, increased bleeding, and macroscopic damages. Pre-treated FJOD animals at both dose levels (2.5 and 5 mL/kg, orally) showed a significant recovery on all these parameters. The colon from the colitis model exhibited a significant increase in the weight/length ratio, edema in submucosa, neutrophil infiltration, necrosis, complete loss of goblet cells, crypt abscess, and gland distortion. While only the FJOD high dose reduced the weight/length ratio alteration, both FJOD doses corrected the other parameters in a dose–response manner. Also, the ulcerative model showed an increase in myeloperoxidase (MPO) and decreased GSH levels in the colon tissue, and increased lactate dehydrogenase (LDH) levels in serum. Once again, the FJOD pre-treatment corrected these values to normal in a dose-dependent way.

Also, one of the selected studies analyzed the gastroprotective effect of betalain-rich ethanol extracts from the fruit (BRE) [45]. To test this *in vivo*, a gastric ulcer Wistar rat model was created by the administration of absolute ethanol at a dose of 0.5 mL/100 g b.w. The results showed that the administration of the extract from the pulp or the peel exerts a dose-dependent protective effect against all major ulcer markers. The more significant values were obtained at the higher dose applied (800 mg/kg b.w.) of peel and pulp extract, showing a reduction in hemorrhagic ulceration (41% and 68%, respectively), and a decrease in the ulcer index (68% and 41%, respectively). Also, the gastric ulceration in the rats induced a significant increase in the volume of gastric secretion as compared to normal rats. Interestingly, the supplementation with pulp and peel extracts at 800 mg/kg b.w. decreased the gastric mucus secretion caused by the ulceration (24% and 35%, respectively). Additionally, the higher dose from the pulp or the peel extracts showed the best antisecretory activity, as evidenced by the rise in the pH by 33% and 44%, respectively. In addition, the extracts from the pulp and the peel decreased LDH activity by 17% and 22% and thiobarbituric acid reactive substances levels by 23% and 41%, respectively. Also, the administration of the extracts from the pulp and peel prevented the gastric mucosal ulcer, the flattening of gastric mucosa, and necrotic lesions in the stomach of rats in a similar way to the well-known gastric protective drug omeprazole, as evidenced by the histological evaluations of gastric damage. Together, all these findings suggest the promising role of BRE from *O. dillenii* in the prevention of gastric damage. This protective effect is in accordance with other similar studies performed by using ethanolic extracts from another cactus such as *O. ficus-indica* [46]. While more studies need to be performed to support the results obtained, these findings highly support the *O. dillenii* as a non-toxic alternative to synthetic treatments against gastric damage, such as omeprazole.

#### 3.6.3. Neuroprotective Effects

Oxidative stress has been pointed out as a major cause of cellular damage in a vast variety of clinical abnormalities including neurodegenerative disorders [47]. Cactus polysaccharides have been reported to display protective effects against H_2_O_2_-induced oxidative injuries [48] and oxygen/glucose deprivation damage [49] in rat brains.

In order to evaluate the neuroprotective effects of ODP, Huang et al. performed a study in an ischemic Sprague-Dawley rat model and characterized the possible mechanisms involved by using a rat pheochromocytoma cell line PC12 [50]. The ischemia rat model was generated by middle cerebral artery occlusion (MCAO) and reperfusion. The intraperitoneal administration of ODP saline solution (200 mg/kg bw) resulted in a significant decrease in cerebral infarct size after 2 h of the ischemia and 24 h reperfusion. Also, the neurological deficit score was inhibited at 6 and 26 h after by ODP treatment. ODP treatment, as shown histologically, markedly attenuated frontal cortex morphological changes, neuronal cell loss, nuclei shrinkage, and dark staining of neurons in the ischemic region.

To further explore the mechanisms behind the *in vivo* results, Huang et al. brought PC12 culture under a 12 h exposure to 0.3 mM H_2_O_2_ [32]. PC12 cells turned out to be very sensitive to H_2_O_2_ damages as shown by a clear decrease in cell viability and increase in apoptosis. However, ODP pre-treatment (0.1, 0.25, 0.5 mg/L) could attenuate H_2_O_2_-induced cell toxicity in a dose-dependent manner. Also, at the higher dose (0.5 mg/L), ODP decreased the percentage of apoptotic cells (from 37.4% to 23.8%) and downregulated, in a dose-dependent manner, the ratio of Bax/Bcl-2 mRNA.

*In vivo* and *in vitro* evidence from this group supports the role of the ODP inhibition of neuronal apoptosis as a main mechanism in neuroprotection. These findings point to the ODP extract from *O. dillenii* as a top candidate for the treatment of oxidative stress-induced neurodegenerative disease. However, further assays should be performed to analyze the mechanisms involved *in vivo*, where the polysaccharide bioavailability will be tested in a more physiological way.

#### 3.6.4. Lung Protective Effects

Epidemiological evidence identifies low-level environmental exposure to Cd as a risk factor for multiple cancers, including lung cancer [51]. A range of 5–50% of inhaled Cd is absorbed through the respiratory tract [52]. To analyze the direct effects of this heavy metal on the lung tissue, Saravani et al. [53] developed an *in vivo* rat model to evaluate cadmium chloride (CdCl_2_)-induced lung toxicity. Rats exposed to CdCl_2_ (2 mg/kg b.w. every 48 h for 16 days) showed congestion, hemorrhage, and structural damage in the bronchi, bronchioles, alveoli, and blood vessels, along with macrophage infiltration indicating inflammation. Pre-treatment with *O. dillenii* hydroalcoholic extract (200 mg/kg b.w.) 90 min before exposure markedly reduced congestion, bleeding, and alveolar irregularities, preserving the basement membrane integrity. Although a star fruit extract (*Averhoa carambola*) was also tested, its protective effect was less pronounced. The study highlighted a strong tissue-protective potential of *O. dillenii* but did not explore the underlying molecular mechanisms.

#### 3.6.5. Nephroprotective Effects

Kidneys are one of the major clearance organs in the human body. In this way, they have a key role in xenobiotic detoxification [54]. Gentamicin is a commonly used antibiotic for Gram-negative bacterial infections, but a major complication associated with its use is nephrotoxicity [55]. To analyze the possible protective effects of ODSO against this unwanted situation, Bouhrim et al. designed an *in vivo* rat model for gentamicin renal damage [56]. The intraperitoneal injection of gentamicin (80 mg/kg b.w./day for 14 days) increased the amount of urea and creatinine, but the co-administration with ODSO significantly counteracted these effects. Also, the antibiotic intake significantly increased the amount of renal gamma-glutamyl transpeptidase (GGT) and the plasma levels of albumin, clear signs of altered renal function. The ODSO attenuated all these alterations. Finally, the histological analysis showed renal corpuscles formed from atrophied glomeruli with expanded Bowman spaces. Once again, concomitant administration with ODSO restored the histopathological insult induced by the gentamicin. Altogether, these results showed that the administration of ODSO and gentamicin simultaneously will be a promising procedure to alleviate the side effects of the antibiotic on the renal tissue. However, as pointed out by the authors, more studies are required to explore the exact mechanisms involved in the ODSO actions against gentamicin-induced physiological disturbances and histopathological alteration in the kidneys.

**Table 5 nutrients-17-03915-t005:** Protective effects of *O. dillenii* against tissular damage.

Biological Model	Treatment	Tissue Damage	Results	Reference
Albino mice	OD fruit extract (0.2 mL by gavage) every 12 h for 7 d after Cd	Hepatic damage Cd (50 ppm in drinking water) for 15 d	↓ Histopathological score	[40]
SPF KM mice	ODP (50–600 mg/kg intragastrically) daily for 7, 14, 21, 28, and 35 d)	Hepatic damage Cd (0.2 mL i.p.) 6 d/w for 21 d	↓ AST, ALT, and ALP Dose- and time- dependent	[41]
Wistar rats	OHAE (100 or 200 mg/kg b.w. orally) once a day for 10 days from 5 d before Pb	Hepatic damage Pb (25 mg/kg b.w./day, i.p.) for 5 d	↓ Histopathological score ↓ AST, ALT and ALP	[42]
Wistar rats	ODSO (2 mL/kg b.w. orally) once a day for 2 w from 7 d before CCl_4_	Hepatic damage CCl_4_ (1 mL/kg b.w. i.p.) once a week for 2 w	↓ AST, ALT, and ALP ↓ Bilirubin direct and total	[29]
Wistar albino rats	FJOD (2.5 or 5 mL/kg, orally once a day for 7 d from 5 d before PCM	Hepatic damage PCM (2 g/kg b.w. orally) in a single	↓ Histopathological score ↓ SGOT, ALP, and SGPT ↓ Bilirubin direct and total Dose-dependent	[43]
Wistar rats	OD peel extract (25 or 100 mg/kg b.w. orally) once a day for 8 w	Hepatic damage HFHF *ad libitum* for 8 w	No significant effects on TG, AST, and ALT	[16]
Wistar rats	OD betalain extract (15, 30, or 60 mg/kg b.w. by gavage) once a day for 3 m	Hepatic damage HFHF *ad libitum* for 3 m	↓ Inflammation ↓ GOT and GPT ↓ Bilirubin direct and total Dose-dependent	[31]
Albino Wistar rats	FJOD (2.5 or 5 mL/kg b.w. orally) once a day for 19 d	Colon damage AcOH (2 mL of 4% *v/v*) intra-rectally on day 8	↓ Colon weight/length ratio ↓ Histopathological score ↓ Bleeding, edema, necrosis ↓ MPO and LDH ↑ GSH	[44]
Wistar rats	BRE from the pulp or peel (200, 400, or 800 mg/kg b.w.) 1 h before absolute ethanol	Gastric ulcerative lesions Absolute ethanol (0.5 mL/100 g b.w. by gavage) in a single dose	↓ Ulcer score ↓ Hemorrhagic ulceration ↓ UI and ↓ VGS at 800 mg/kg dose	[45]
Sprague-Dawley rats	ODP (200 mg/kg b.w. i.p.) at 72, 48, and 24 h before MCAO and 15 min after MCAO	Cerebral ischemia MCAO with reperfusion after 2 h	↓ Cerebral infarct size ↓ Neurological deficit score ↓ Histopathological score	[50]
Wistar rats	OD hydroalcoholic extract (200 mg/kg b.w. by gavage) every 48 h, 90 min before the Cd, for 16 d	Lung damage Cd (2 mg/kg b.w. By gavage) every 48 h for 16 d	↓ Congestion and bleeding ↓ Alveolar and airway irregularities Intact basement membrane	[53]
Wistar rats	ODSO (2 mL/kg b.w. by gavage) 3 h before the Gentamicin for 14 d	Renal damage Gentamicin (80 mg/kg b.w. i.p.) for 14 d	↓ Urea and creatinine ↓ GGT ↓ Albumin ↓ Histopathological score	[56]

AcOH: Acetic acid; ALP: Alkaline phosphatase; ALT: Alanine aminotransferase; AST: Aspartate aminotransferase; BRE: Betalain-rich ethanol extract; b.w.: Body weight; FJOD: Fresh fruit juice of *O. dillenii*; GGT: Gamma-glutamyl transpeptidase; GOT: Glutamic-oxaloacetic transaminase; GPT: Glutamic pyruvic transaminase; GSH: Glutathione; HFHF: High-fat high-fructose diet; i.p.: intraperitoneal; LDH: Lactate dehydrogenase; MCAO: Middle cerebral artery occlusion; MPO: Myeloperoxidase; OHAE: *O. dillenii* hydroalcoholic extract; OD: *O. dillenii*; ODP: *O. dillenii* polysaccharide; ODSO: *O. dillenii* seed oil; PCM: Paracetamol; SGOT: Glutamic-oxaloacetic transaminase; SGPT: Serum glutamic pyruvic transaminase; TG: Triglyceride; UI: Ulcer index; VGS: Volume of gastric secretion. The arrows (up or down) indicate increases or decreases in biomarkers resulting from the administration of *O. dillenii*.

It is evident that the preventive effects of *O. dillenii* on cellular and tissue damage are a common element in all studies. However, the results described should be taken with caution, since in most cases only a single study is available for each tissue and the doses and routes of administration are not compatible with its use as food or nutraceuticals. Thus, it will be interesting to focus future experiments on elucidating the mechanisms responsible for these protective actions. These assays should use physiological doses and routes of administration to ensure stronger and consistent results that could be translated to human clinical trials.

### 3.7. Antimicrobial Activity

The antimicrobial efficacy of *O. dillenii* has been the subject of seven *in vitro* works compiled in this review [57,58,59,60,61,62,63]. Different extracts from different parts of the plant were tested against several bacterial strains; the most studied were *Staphylococcus aureus* [57,58,59,61,62,63], *Escherichia coli* [57,58,59,60,61,62], and *Bacillus subtilis* [57,58,60,62].

Growth inhibition of *S. aureus* was tested using aqueous whole *O. dillenii* plant extract [57], aqueous and ethanolic extracts from the cladodes [58], different solvent extracts from the flowers [59], ethanolic extracts from the seeds, peel, and juice [61], and non-polar (petroleum and chloroform) and polar (methanol and water) extracts from the stems [62]. Almost all the extracts showed some inhibitory effect on the growth of this Gram-positive bacteria, the only negative data being the one from the Loukili et al. [61] study, related to the seeds, skin, and juice of the fruit. Interestingly, one work also explored the effects of the ethanolic extract from the leaves of *O. dillenii* on methicillin-resistant *S. aureus* isolates from nosocomial infections, showing a very low activity against this antibiotic resistant strain [63]. *B. subtilis* seemed to be sensitive to some of the *O. dillenii* extracts, such as the aqueous whole plant one [57], the aqueous seed one [60], and the petroleum and chloroform stems ones [62]. Despite that, the cladodes showed a non-significant effect in comparison to the one shown for other Gram-positive strains such as *S. aureus* [58].

All *O. dillenii* extracts tested for their bactericide role against *E. coli* showed a not significant good effect, independently of the part of the plant, or the solvent used [57,58,59,60,61,62]. Only the ethanolic whole fruit extract seemed to exhibit a weak to moderate activity against this species [61]. For *Salmonella typhi*, also a Gram-negative bacteria, the intervention with both non-polar (petroleum ether and chloroform) and polar (methanol and water) extracts from the dried stems did not provide a considerably good antimicrobial effect [62]. These results are supported by the findings of Ahmed et al. [57] which showed a very low effect of the whole plant aqueous extract. In the case of another Gram-negative bacteria such as *Klebsiella pneumonia*, skin water extract and seed ethanolic extract from *O. dillenii* have low minimum inhibitory concentration (MIC) values [60] but no extract (ethanolic or aqueous) of the cladodes exhibited significant antibacterial properties [58]. Despite that, the ethanolic extract from *O. dillenii* cladodes showed a moderate antimicrobial effect against Gram-positive *Micrococcus luteus* while the aqueous extract did not show inhibitory effects at all. Interestingly, the inhibitory activity for both polar extracts was not significant in the case of *B. cereus* and *Salmonella enteritidis* [58]. While the aqueous extract of the whole *O. dillenii* plant showed a truly discrete bactericidal effect on the *Pseudomonas aeruginosa* [57], the flower extract exhibited promising results on the inhibition of this strain [59]. Despite the already described promising effects of the flower extracts against *S. aureus* and *Pseudomonas aeruginosa*, in the case of *Enterococcus hirae* they showed a very low antibacterial activity [59]. Ethanol and aqueous extracts from the seeds and diethyl ether extracts from the juice from *O. dillenii* showed the lowest MIC values in the treatment of *Pseudomonas fluorescens* while their effects against *Micrococcus lysodeikticus* and *Enterococcus faecalis* were not significant in comparison [60]. Finally, the work of Loukili et al. [61] showed a weak to moderate antibacterial activity of the ethanolic extract from the juice of *O.dillenii* against *Listeria monocytogenes*, and *Salmonella braenderup*, while seeds and peel extracts showed no effect at all.

All these results suggest a trend that Gram-positive bacteria could be more sensitive to the antimicrobial properties of the cladodes extracts than the Gram-negative ones, something that Lataief et al. [58] already highlight in their work. Nevertheless, there seem to exist some exceptions such as the non-inhibitory effects against *S. aureus* for the seeds, skin, and juice extracts obtained by Loukili et al. [61] or the lack of activity of the cladodes ethanolic extract against *B. subtilis* [58] or the surprising antimicrobial effect of the flower extract against the Gram-negative *P. aeruginosa* [59]. Despite the MIC and/or diameter of the inhibition area being used in all of the studies, the different concentrations and types of antibiotic used as standard clearly make the direct comparison among the different results shown difficult. Also, the various parts from the *O. dillenii* and the several methods of extraction seem to have a clear influence on the antimicrobial activities. There is a wide range of molecules with antimicrobial potential in the plant such as the quinic acid [58], phenolic compounds (especially flavonoids) [64], d-limonene [65], or catechin and gallic acid [61], so the quantity of them in the different parts and the concentration obtained by the different solvents seem to be a crucial factor for the inhibitory activities. Additionally, all the studies were perfomed *in vitro* so the antimicrobial activity of *O. dillenii* must be tested *in vivo* in order to describe the exact molecules responsible for the observed effects, as well as their mechanisms in action.

### 3.8. Antifungal Activity

Four of the previously cited studies also extended their research to the *O. dillenii* antifungal activity. Once again, different extracts from different parts of the plant were tested against several fungal strains.

Aqueous whole *O. dillenii* plant extract seems to exhibit only mild antifungal activity against *Candida albicans*, the most common human fungal pathogen, *Aspergillus niger*, or *Penicillum notarium* [57]. Aqueous and ethanolic extracts from the cladodes [58] showed no antifungal effect against *Fusarium* sp. and *Pythium catenulatum* but, interestingly, showed a very good antimicrobial activity (similar to the cycloheximide antibiotic control) against *Fusarium oxysporum*, independently of the solvent used.

In the Katanic et al. [60] work, the extracts of *O. dillenii* seeds from the Nador locality showed the lowest overall antifungal MIC values, and for the seeds of the plant material from Essaouira, only ether and aqueous extracts were effective on six fungal species (*Trichoderma harzianum*, *Penicillum cyclopium*, *Aspergillus niger*, *Doratomyces stemonitis*, *Phialophora fastigiata*, and *Fusarium oxysporum*). The most resistant species in the evaluation of antifungal activity was yeast *Candida albicans*. Moreover, almost all skin extracts tested, except the aqueous ones, were inefficient at the highest concentration used.

Finally, Loukili et al. [61] tested the antifungal properties of the seeds, skin, and juice of the fruit, but their results showed no effect against *Candida prusei*, *Candida albicans*, *Candida tropicalis*, or *Saccharomyces cerevisiae.*

It is clear that the antifungal properties of *O. dillenii* are lower than its antibacterial activities. Their capacity against most of the fungal strains were mild or even inexistent in the majority of cases with only some exceptions against *Fusarium oxysporum* [58,60] and the ether and aqueous extracts from the Essaouira seeds that had positive results against six fungal species demonstrated only by one group [60]. As in the antimicrobial studies, all the experiments were perfomed *in vitro* so all the evidence must be verified *in vivo* to clearly describe the *O. dillenii* behavior in a physiological environment.

Once again, the different concentrations and types of antibiotics used as standard clearly hinder direct comparison among the reported results. Also, the use of different plant parts, extraction solvents, and experimental dosages further complicates comparison across the related studies.

### 3.9. Antiviral Activity

Only one study focused on the antiviral capacity of *O. dillenii* [66]. In this case, Kumar et al. tested the potential of the methanolic extract from the flowers against thirteen different virus strains. In the antiviral assay, based on the cytopathic effect (CPE) of the virus infected monolayer of the different cells, the extract showed the strongest antiviral activity against herpes simplex 1 (EC50 = 25 µg/mL) and 2 (EC50 = 20 µg/mL), and moderated activity against vaccinia (EC50 = 100 µg/mL). The extract was not toxic against the rest of the virus strains tested: Vesicular stomatitis, Feline corona, Feline herpes, Coxsackie B4, Respiratory syncytical, Parainfluenza-3, Reo-1, Sindbis, Coxsackie, and Punta toro (EC50 > 100 µg/mL). These results point to the possible utility of *O. dillenii* against some virus infections, as a direct antiviral or in combination with antiviral therapy, to reduce the doses of the drug treatment and then its secondary effects. Nevertheless, only one study is not enough, and more *in vitro* and *in vivo* research is needed in order to corroborate and consolidate the antiviral capacities of *O. dillenii*. In addition, we must be cautious about the antiviral properties of the fruit as it did not exhibit significant effect on many virus strains.

## 4. Conclusions, Limitations, and Future Perspectives

The evidence compiled in this systematic review underscores the wide-ranging biological potential of *O. dillenii*, including its anti-inflammatory, lipid-modulating, cardioprotective, antidiabetic, antiproliferative, tissue-protective, and antimicrobial activities. Studies consistently demonstrate that extracts and isolated compounds can modulate key proliferative and inflammatory pathways, improving lipid and glucose metabolism and protecting tissues against diverse toxic or metabolic insults. Collectively, these findings position *O. dillenii* as a promising nutraceutical candidate for the prevention and complementary management of NCDs.

Nevertheless, significant methodological and translational limitations remain. A substantial number of studies rely on *in vitro* assays, which, while informative for mechanistic understanding, often overlook essential physiological processes such as gastrointestinal digestion, intestinal absorption, hepatic metabolism, and systemic bioavailability. The direct exposure of cell lines to crude extracts or isolated compounds at non-physiological concentrations limits the extrapolation of these findings to human conditions. To enhance the physiological relevance of cellular studies, future research should incorporate *in vitro* digestion and bioaccessibility models, followed by exposure to intestinal barrier systems (e.g., Caco-2 or co-culture models) or gut–liver axis platforms that simulate first-pass metabolism and transport. These approaches would allow a more accurate estimation of the fraction of bioactive compounds reaching target tissues and their metabolic transformations.

Results from *in vivo* models are generally consistent and indicate a wide range of tissue protection, mainly on the liver, and metabolic benefits. However, the studies exhibited significant heterogeneity in several key aspects, such as study design (cell types and animal models), plant parts used (cladodes, fruit, seeds, peel or pulp), plant fractions (methanolic, aqueous), extraction methods (chromatography, ultrasound-assisted), doses, routes of administration, and treatment durations. In addition, few investigations have elucidated the molecular pathways mediating these effects, and the identification of active compounds remains incomplete in several cases. Future preclinical works should therefore combine biochemical, histological, and omics-based approaches to clarify mechanistic targets and synergistic interactions among phytochemical classes.

Additionally, essential data for quantitative synthesis, such as means, standard deviations, and confidence intervals, were either missing or incompletely reported, which limits the ability to make robust comparisons. Another important limitation identified is the frequent presence of high or unclear risk of bias in the primary studies, which stems from inadequate or incomplete reporting. The biases detected always appear in the section referring to performance, detection, attrition, and reporting, demonstrating that studies have not acknowledged the need to apply blinding at these stages of the process. This lack of transparency compromises the reliability of the findings and makes it difficult to assess the true effects of the interventions. Therefore, future research should focus on standardizing these factors to allow for more meaningful comparisons and clearer mechanistic interpretations.

The absence of clinical evidence constitutes the major gap in the current body of research. No controlled human studies have yet confirmed the metabolic, anti-inflammatory, or tissue-protective effects observed in animal models. Rigorous clinical trials are required, initially in healthy volunteers, to assess safety, tolerability, and preventive efficacy. Then, these should be followed by intervention studies in patients with metabolic syndrome, dyslipidemia, or inflammatory disease to explore potential adjuvant roles. Such studies should also define pharmacokinetic parameters, optimal dosage forms, and long-term safety profiles. Moreover, there is a clear need for human studies to assess the effectiveness of these compounds in various dietary forms, such as powdered supplements or juices, in order to evaluate their bioavailability and overall efficacy. It is also crucial to determine whether typical dietary intake can provide the physiological doses necessary for potential therapeutic effects. Furthermore, research should focus on exploring encapsulation techniques and other advanced delivery systems, as these may enhance the bioavailability, stability, and controlled release of the compounds, ultimately improving their therapeutic potential. In this sense, the high doses used in animal studies may not directly reflect realistic human dietary intakes. However, nutraceutical formulations such as concentrated extracts, powders, or encapsulated forms present a feasible approach to achieving these higher doses in humans. These formulations allow for the precise and controlled delivery of bioactive compounds, potentially enabling individuals to reach the doses that have demonstrated efficacy in preclinical studies. Therefore, further research is necessary to identify the optimal dosages for human consumption and to evaluate the safety, bioavailability, and effectiveness of these formulations in clinical settings.

Additionally, despite the promising therapeutic potential of *O. dillenii*, several intrinsic challenges limit its broader application. These include significant variability in the concentration of bioactive compounds across different plant parts, as well as environmental factors that can also influence its chemical composition. Such inconsistencies complicate the efforts to standardize its use in clinical or commercial settings. Addressing these issues will require more rigorous and well-controlled studies to ensure the efficiency of this plant.

In summary, *O. dillenii* represents a multifunctional nutraceutical resource with considerable preventive and therapeutic activities. Future research should prioritize physiologically relevant experimental designs, standardized extraction and dosing protocols, and comprehensive clinical validation. Only through this integrative approach, from digestion and absorption models to human trials, can the translational potential of *O. dillenii* be fully described for evidence-based nutritional and medical applications.

## Figures and Tables

**Figure 1 nutrients-17-03915-f001:**
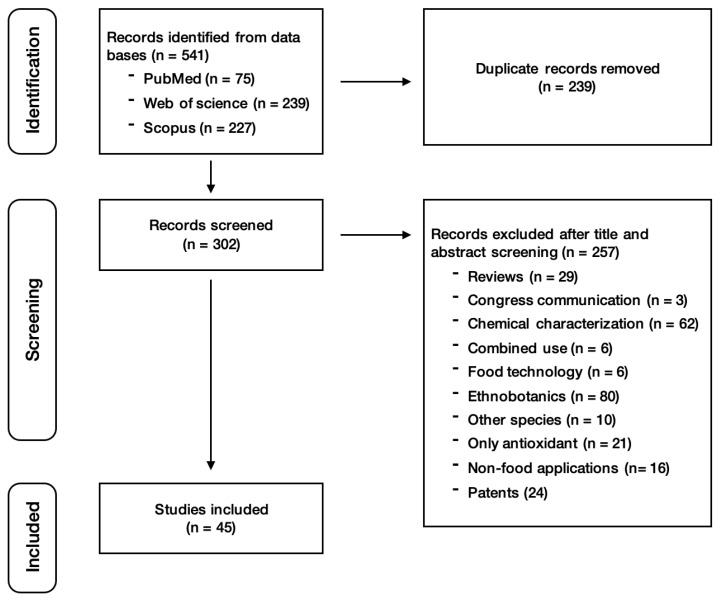
Search strategy for primary studies.

**Figure 2 nutrients-17-03915-f002:**
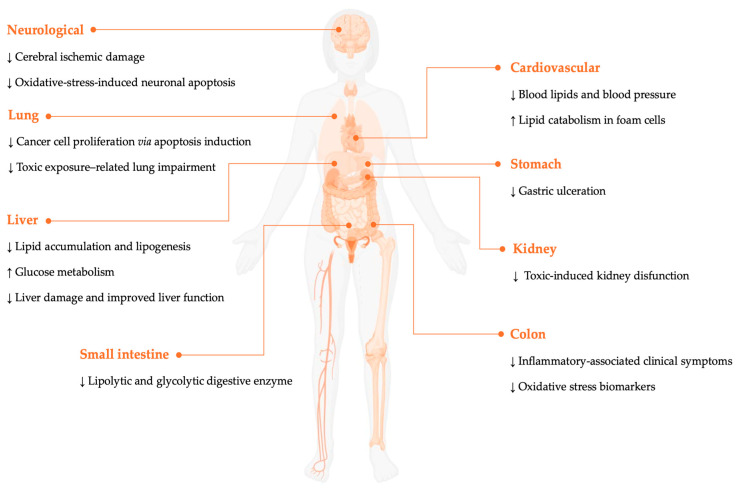
Preventive and therapeutic actions of *O. dillenii.* The arrows (up or down) indicate increases or decreases in the biomarkers after the administration of *O. dillenii*.

**Table 2 nutrients-17-03915-t002:** Comparison of antiproliferative effects.

Biological Model	Treatment	Assay	Results	Reference
293/NF-κB-Luc HEK cells	6 h with MeOH fruit extract (20 μg/mL)	Inhibition of TNF-α-induced NF-κB	18.2% inhibition	[12]
**-**	30′ at 37 °C before quenching with NaOH	Aromatase inhibition assay	12.5% inhibition	
COS-1 cells	24 h with fruit extract	RXR/luciferase assay	No significant results	
Hepa 1c1c7 cells	48 h with fruit extract (20 μg/mL)	QR1 assay	No significant results	
MCF-7/LU-1/MD-MB-231 cells	72 h with fruit extract	SRB assay	No significant results	
SK-MES-1 cells	24–48 h with cactus polysaccharides. (0.005625–1.44 mg/mL)	Cell proliferation assay	70% inhibition rate at 1.44 mg/mL 48 h	[20]
	24–48 h with cactus polysaccharides (0.18–0.36 mg/mL)	Morphology test	Dose-dependent dispersion and fuzzy	
	24 h with cactus polysaccharides (0, 0.18 to 0.36 mg/mL)	Cell cycle arrest and apoptosis	Cells blocked in S phase, ↑ apoptosis, p53, and PTEN expression	
EAC cells	48 h with betacyanin, pectin or pulp	Apoptosis	↑ Apoptosis: Betacyanin (78.8%)	[21]
CAM	Cladodes aqueous extract (4–10%)	Macroscopic evaluation	↓ Blood vessels diameter	[22]
		3D topographical evaluation	Deteriorated angular spectrum and reduction in the height of blood vessels	
		Histological evaluation	Thinning of ectodermal layer and damaged extracellular matrix	

CAM: Chicken chorioallantoic membrane; EAC: Erlich ascites carcinoma; MeOH: Methanol; NF-κB: Nuclear factor kappa-light-chain-enhancer of activated B cells; PTEN: Phosphatase and tension homolog deleted on chromosome ten; QR1: Quinone reductase-1; RXR: Retinoide X receptor; SRB: Sulforhodamine B; TNF: Tumor necrosis factor.

**Table 4 nutrients-17-03915-t004:** Comparison of antidiabetic effects.

Biological Model	Treatment	Assay	Results	Reference
New Zealand rabbits	Fresh filtered juice alloxan-induced DMII	Serum biochemical determination	↓ Hyperglycemia =Plasma insulin	[32]
Sprague-Dawley rats	Fruit pad and pulp extracts	Sugar blood levels	↓ Fasting blood glucose levels	[33]
Chinese Kunming mice	ODP-Ia 100–400 mg kg/b.w. (22 days)	Serum biochemical determination	↓ Blood glucose =Plasma insulin ↑ b.w. ↑ Hepatic glycogen	[34]
Sprague-Dawley rats	ODFP STZ-induced DMII	Serum biochemical determination	↓ Blood glucose ↓ Excessive food and water consumption ↓ Urine production ↑ b.w.	[35]
		Histological examination	Improve the structure integrity of pancreatic islet	
Wistar rats	ODSO (1–2 mL/kg/day)	Serum biochemical determination	↓ Fasting blood glucose, glycosuria, TC, TG, AST and ALT ↑ Hepatic glycogen ↑ Food intake ↓ b.w. loss and urinary volume	[36]
Wistar rats	Betanin from pulp (60 mg/kg b.w.) (DMII HFHF-induced)	Serum biochemical determination	↓ Blood sugar ↓ Intestinal α-amylase, maltase, and sucrase ↑ Glucokinase, glycogen ↓ Glucose-6-phosphatase	[31]

b.w.: Body weight; HFHF: High-fat high-fructose diet; ODP: *Opuntia dillenii* polysaccharide; ODSO: *Opuntia dillenii* seed oil; ODFP: *Opuntia* polysaccharide. The arrows (up or down) indicate increases or decreases in biomarkers resulting from the administration of *O. dillenii*.

## Data Availability

No new data were created or analyzed in this study. Data sharing is not applicable to this article.

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
