# Peer review of "Opuntia dillenii as a Nutraceutical and Dietary Resource for Disease Prevention and Management: A Systematic Review"

_nutrients, 2025, doi:10.3390/nu17243915_

Round 1
Reviewer 1 Report
Comments and Suggestions for Authors
The attached systematic review examines the potential of Opuntia dillenii as a nutraceutical and dietary resource for disease prevention and management, with a focus on non-communicable diseases (NCDs) driven by inflammation and metabolic dysfunction. This review follows PRISMA 2020 guidelines with clear protocol details for literature search and screening, using PubMed, Scopus, and Web of Science, and precise inclusion and exclusion criteria. Despite its systematic nature, the review faces several challenges.
Specific comments:
- Most included studies are preclinical (in vitro or animal), with few clinical trials, limiting its translational value. The review rightly emphasizes the need for well-designed human clinical studies validating efficacy, safety, and pharmacokinetics.
- Heterogeneity in extraction methods, compound characterization, and bioactivity assays, and absence of standardization complicate comparison and mechanistic interpretation.
- Risk-of-bias tables indicate frequent “unclear” or “high risk,” reflecting poor reporting across primary studies.
- Selective exclusion of antioxidant-only studies may neglect insights into how antioxidant and other properties interact.
- Some sections display descriptive rather than critical synthesis, particularly for studies lacking statistically significant effects; mechanistic gaps and contradictory results (e.g., in anti-inflammatory assays) could be discussed in more depth.
Reviewer 2 Report
Comments and Suggestions for Authors
The aim of this review was to review the scientific literature on the properties of Opuntia dillenii. The literature review should further describe the origin and properties of the cactus.
Were online sources also included in the review?
Please describe the threats and opportunities posed by the described plant in greater detail. It is recommended to use diagrams or graphs illustrating the data in literature reviews.
Reviewer 3 Report
Comments and Suggestions for Authors
This work addresses an increasingly relevant topic in nutritional science and functional food research, especially given the rising interest in plant-based compounds for metabolic and chronic disease modulation. The manuscript is aligned with the thematic scope of Nutrients and demonstrates substantial effort in systematically collecting and summarizing current evidence related to Opuntia dillenii.
The review presents an extensive compilation of in vitro and in vivo findings and provides a broad overview of the potential biological effects of O. dillenii, including anti-inflammatory, antidiabetic, lipid-modulating, hepatoprotective, and antiproliferative actions. The systematic approach guided by the PRISMA framework strengthens the methodological rigor and improves transparency.
However, while the manuscript is rich in data, it currently reads as a descriptive catalogue of results rather than a critical synthesis of evidence. Several areas require strengthening — particularly the structure of the discussion, clarity and quality of English writing, and deeper evaluation of the strength and limitations of existing studies.
The topic is timely and underrepresented in current scientific literature, particularly regarding Opuntia dillenii compared to the more commonly studied O. ficus-indica.
Use of PRISMA guidelines and multiple scientific databases increases reliability and reproducibility.
The manuscript covers a wide range of biological functions and bioactive components.
Tables summarizing findings allow quick comparison between studies and represent a useful reference resource for researchers.
The manuscript would benefit from thorough English language editing. Currently, syntax, punctuation, and grammar interfere with clarity. Several paragraphs contain overly long sentences, repeated expressions, or inconsistent terminology. Improving writing flow will make the text more accessible and professional.
Although the evidence is well reported, the manuscript lacks a critical evaluation of:
-
methodological rigor of included studies,
-
reproducibility and consistency of findings,
-
potential publication or reporting bias,
-
safety considerations and toxicity,
-
and limitations arising from the predominance of animal and in vitro models.
A dedicated subsection discussing these aspects would significantly elevate the scientific value of the paper.
The review highlights promising preclinical data, yet no human clinical trials are available. The conclusions should therefore remain cautious and emphasize the early stage of the evidence. Additionally, tested doses in animal studies are often high and may not reflect realistic dietary or supplement intakes. A short discussion comparing experimental dosages with potential human intake levels would strengthen the translational context.
In its current form, the Discussion section largely repeats results. Instead, it should synthesize trends, identify which biological effects are strongly vs. weakly supported, highlight mechanistic patterns, and point out inconsistencies. A graphical summary or evidence-strength grading table could be considered.
Moreover:
-
Botanical names should be italicized consistently (Opuntia dillenii).
-
Abbreviations should be defined once and listed or used consistently.
-
Figure legends should clearly define all abbreviations.
-
A justification for not performing a meta-analysis should be added.
-
Ensure that reference style follows Nutrients guidelines.
At this stage, the manuscript contains valuable scientific content and is suitable in scope for the journal; however, significant revisions are needed to improve clarity, coherence, and critical depth. Once the concerns regarding writing quality, analytical depth, and interpretative strength are addressed, the manuscript has strong potential for publication.
Comments on the Quality of English LanguageClarity and quality of English writing should be improved.
Round 2
Reviewer 2 Report
Comments and Suggestions for Authors
The manuscript has been revised based on the reviewer's suggestions and concerns. It may be subject to the editor's discretion.
Author Response
Dear Reviewer,
Thank you very much for your careful reading of our manuscript and for your constructive and detailed comments. We greatly appreciate the time and expertise you devoted to improving the quality and clarity of our work.
Following your suggestions, we have thoroughly revised the text to improve the English language, including grammar, spelling, and overall style. We have also worked on enhancing the fluency and coherence of the manuscript, refining sentence structure, improving transitions between sections, and ensuring consistent use of terminology throughout.
We are confident that these changes have strengthened the manuscript, and we remain at your disposal for any further clarifications or additional modifications you may consider necessary.
Kind regards,
Nisa Buset-Ríos
on behalf of all co-authors
Reviewer 3 Report
Comments and Suggestions for Authors
The revised version of the paper is suitable for publication.
Comments on the Quality of English LanguageClarity and quality of English writing should be improved.
Author Response
Dear Reviewer,
We would like to sincerely thank you for your thorough evaluation of our manuscript and for the insightful comments you provided. Your observations have been very valuable in helping us to improve the overall quality of the work.
In response to your suggestions, we have carefully revised the entire manuscript to refine the English language, correcting grammar and spelling and improving sentence structure. We have also focused on enhancing textual fluency and coherence, clarifying ambiguous passages, smoothing transitions between sections, and ensuring consistent use of scientific terminology.
We believe that these revisions have substantially strengthened the manuscript, and we remain at your disposal for any further comments or adjustments you may consider appropriate.
Kind regards,
Nisa Buset-Ríos
on behalf of all co-authors